# Screening and Optimization of Demulsifiers and Flocculants Based on ASP Flooding-Produced Water

**Bin Huang [1,\*], Jie Wang [1], Wei Zhang [3], Cheng Fu [1,2,\*], Ying Wang [4] and Xiangbin Liu [5]**

[1]   Key Laboratory of Enhanced Oil Recovery, Ministry of Education, College of Petroleum Engineering, Northeast Petroleum University, Daqing 163318, China; wjdar123@163.com

[2]   Post-Doctoral Scientific Research Station, Daqing Oilfield Company, Daqing 163413, China

[3]   Shandong Key Laboratory of Oil-Gas Storage and Transportation Safety, College of Pipeline and Civil Engineering, China University of Petroleum, Qingdao 266580, China; upc_NGHzw@163.com

[4]   Aramco Asia, Beijing 100102, China; ying.wang.1@aramcoasia.com

[5]   Research Institute of Production Engineering, Daqing Oilfield, Daqing 163453, China; liuxiangbin@petrochina.com.cn

\*   Correspondence: huangbin111@163.com (B.H.); cheng_fu111@163.com (C.F.)

**Abstract:** The water produced by alkaline–surfactant–polymer (ASP) flooding is difficult to treat due to the presence of residual chemicals. Therefore, research and development of efficient and low-cost methods for the treatment of ASP flooding produced water is necessary. Chemical destabilization is the most common and effective way to treat the produced water. This paper describes an optimization method for demulsification and flocculation. Some demulsifiers and flocculants commonly used in oilfields were screened, compounded, and optimized. Since the effect of treatment using only demulsifier or flocculant to treat the produced water is often not enough to meet the reinjection standard, an orthogonal experiment was carried out to study the demulsification–flocculation method to treat produced water. Five main influencing factors of the oil concentration were investigated. Based on the results of the range analysis and the relationship between the five factors and oil concentration, the order of significant factors was found to be demulsifier dosage > flocculant dosage > settling time > stirring time > stirring intensity, and the optimal demulsification–flocculation treatment conditions were successfully optimized.

**Keywords:** emulsifier; flocculant; produced water; ASP flooding; orthogonal experiment

## 1. Introduction

ASP flooding (a methodology of alkaline–surfactant–polymer flooding), as one of the most effective oil displacement technologies, has been popularized in most oilfields because it can increase the oil recovery rate by 20% compared with water flooding [1]. However, as the largest by-product of the oilfield, ASP flooding produced water increases year by year. In China, $5 \times 10^8$ m$^3$ of the produced water by oilfields needs to be treated per year. In order to avoid environmental damage caused by secondary pollution, the development of efficient water treatment technology is one of the key problems that needs to be solved urgently at the present stage of oilfield development [2].

Due to the addition of alkali, surfactant and polymer, which compete with the traditional produced water, ASP flooding produced water has characteristics of strong emulsification stability, high interfacial membrane strength, small oil droplet size, phase stability and high salinity [3–7]. Furthermore, if the produced water is directly discharged into the environment, it will pollute the water environment and the soil environment, causing blockage of the bottom layer and affecting human health [8]. Therefore, ASP flooding-produced water treatment has become an important factor restricting the development of ASP flooding technology.

Since 2012, a sequential batch sedimentation treatment process was proposed for the treatment of water produced in the Daqing oilfield, which takes into account the synergistic effects of physical chemistry [9]. This method adopts settlement equipment with a long settling time and impact resistance as well as a filter tank which can reduce the filtration rate. When the ions of the ASP flooding produced water reach a state of supersaturation, a water stabilizer is added to meet the water quality requirements of ASP flooding produced water. However, in the actual operation process, problems such as leaking points of the aeration pipeline and perforation of the heating coil were also found, which affected the final treatment efficiency [9]. Di Mao et al. [10] studied the air floating/hydrolysis acidification/biological contact oxidation/two-stage sand filtration process for the treatment of ASP flooding-produced water. After field commissioning, under optimal conditions, the oil removal rate reached up to 98% and the suspension removal rate reached up to 95%, which can ensure that the water produced reaches the "double 20" injection water quality standard of a high-permeability reservoir. Furthermore, the air floatation method generates bubbles by injecting air into the water, and relies on the characteristics of the bubble surface to adsorb oil droplets or suspended matter to achieve separation [11–13]. Jiang Limin [14] invented a carbon dioxide dissolved gas device. When produced water flows into the water tank, $CO_2$ in the carbon dioxide tank is mixed with the return water in the return pipe through the dissolved gas pump. The mixture then passes through the dissolved gas release device on the outlet pipe to produced water for carbon dioxide flotation treatment. After the dissolved gas treatment, the ASP flooding produced water will bring the crude oil and the floccus to the liquid surface under the action of carbon dioxide bubbles, which is then collected by a scraper to remove the emulsified oil and the suspended matter. Thereby, the removal of emulsified oil and suspended matter is achieved.

Although the treatment processes listed above can enable the treated ASP flooding produced water to reach the standard of oilfield reinjection water, each process has its own shortcomings. The filtration combined process has the problem of high operating cost. The air floating/hydrolysis acidification/biological contact oxidation/sand filtration process requires multiple additions of high-efficiency microbial agents. In the dissolving air flotation process, water quality regulators and compounding agents need to be added. Water quality regulators and compound agents are also needed in the operation of the dissolved gas flotation process. This is mainly due to the presence of residual chemicals in the produced water, resulting in serious emulsification of the produced water, making the oil and water difficult to separate [15–17]. Therefore, the chemical destabilization of oil–water emulsion is the key to solving this problem.

The addition of demulsifiers and flocculants is still the main means to destabilize ASP flooding-produced water in an oilfield. The combination of different agents and the optimization of their conditions are the main methods used to develop the optimum treatment. Therefore, this paper studies some demulsifiers and flocculants commonly used in oilfields. The materials were screened, compounded, and optimized. Then, an orthogonal experiment was carried out to study the demulsification–flocculation method to treat the produced water. Based on the results of the range analysis and the relationship between five influencing factors and the oil concentration, the optimal demulsification–flocculation treatment conditions were optimized, and the optimized approach was shown to be an effective method for the treatment of ASP flooding-produced water. By studying the screening and optimization method of demulsification and flocculation for the treatment of ASP flooding produced water, we aim to provide a method for screening and optimization of chemical reagent for the chemical treatment to deal with the ASP flooding-produced water, so that the ASP flooding-produced water can be effectively treated.

## 2. Materials and Methods

### 2.1. Materials

The crude oil for this study was obtained from the No.1 Oil Production Company in Daqing, with a water concentration of less than 0.5%. The viscosity of crude oil is 60 mPa·s and its density is 860 kg/m$^3$ at 45 °C. The polymer (partly hydrolyzed polyacrylamide, HPAM) was supplied by Daqing Oilfield Production Technology Institute, with an average molecular weight (MW) of $3.0 \times 10^6$ and a degree of hydrolysis of about 25–30%. The surfactant ORS-41 (the major component of which is alkylbenzene sulphonate) was supplied by the No.1 Oil Production Company in Daqing, and the active concentration is 50 wt%. The alkali used in this study was NaOH, which provided by Daqing Oilfield Production Technology Institute, with a concentration of 30%. This was used as an analytical reagent. Demulsifiers and flocculants were purchased from Daqing Oilfield Production Technology Institute.

### 2.2. Preparation of Simulated Produced Water

According to the actual underground water qualities in the Daqing oilfield, the mineralized water was prepared first. The salts contained in the mineralized water were shown in Table 1 (expressed as mg/L).

**Table 1.** Composition of mineralized water.

| Composition | NaCl | NaHCO$_3$ | Na$_2$CO$_3$ | Na$_2$SO$_4$ | CaCl$_2$ | MgCl$_2$·6H$_2$O |
|---|---|---|---|---|---|---|
| Concentration (mg/L) | 1523 | 2820 | 168.7 | 10.5 | 56.9 | 35.5 |

After that, the mineralized water was used to prepare simulated ASP flooding produced water. The method for preparing the produced water in the lab was as follows: 200 g of mineralized water with 1000 mg/L of HPAM and 1000 mg/L of ORS-41was added to a 500 mL beaker and heated to 45 °C in a water bath for a period of time. Then 200 g of crude oil was added to the beaker and the mixture was heated to 45 °C in the water bath for 1 h. After that the mixture was emulsified for 5 min at 20,000 rpm with an emulsifier to obtain a 50% oil-in-water emulsion. Next, 199.2 mL of mineralized water with different concentrations of alkaline, surfactant, and polymer was prepared in a 250 mL beaker, and 0.8 g of the 50% oil-in-water emulsion was added to the solution. The sample was shaken to produce simulated produced water with an oil concentration of 2000 mg/L.

### 2.3. Oil Concentration Determination

In this paper, the oil concentration determination was performed by SY/T 5329-2012 "water quality standard and practice for analysis of oilfield injecting waters in clastic reservoirs". First, 200 g of the produced water was prepared in a 250 mL beaker and allowed to settle for 4 h at 45 °C in an oven. Then 50 mL of the water was taken from the bottom of the beaker using a syringe to determine its oil concentration.

### 2.4. Screening Tests of Demulsifiers

In this paper, different types of water-soluble demulsifiers and oil-soluble demulsifiers (purchased from Daqing Oilfield Production Technology Institute, the purity of demulsifiers is about 95%) commonly used in oilfields were screened. The water-soluble demulsifiers included: DPR-1193, DPR-1612, DPR-1814, DPR-1815, DPR-1884, and DPR-1885. The oil-soluble demulsifiers included: DYRPR-1625, DYRPR-1712, DYRPR-1881, DYRPR-1882, DYRPR-1869, and DYRPR-1870.

The bottle-test method was used to screen the demulsifiers. The demulsifiers were added in concentrations of 100 mg/L and 200 mg/L, stirred at 200 rpm for 2 min, and settled in a 45 °C water bath for 2 h. The oil concentration in the aqueous phase of the sample before and after the treatment was detected.

## 2.5. Screening Tests of Flocculants

The flocculants used in this paper included inorganic flocculants and organic flocculants (purchased from Daqing Oilfield Production Technology Institute, the purity of flocculants is about 28–30%). The inorganic flocculants were polyaluminum chloride (PAC), polyferric sulfate (PFS), and polymeric aluminum ferric chloride (PAFC). The organic flocculants were cationic polyacrylamide (CPAM, MW = $3.2 \times 10^6$), anionic polyacrylamide (APAM), polyacrylamide (PAM), D2N-1647, D2N-1651, D2N-1650, and D2N-1648. All of the flocculants are industrial products.

In these experiments, inorganic flocculants at a concentration of 200 mg/L and organic flocculants at a concentration of 10 mg/L were added to the produced water. The solution was stirred at 200 rpm for 2 min with a mixer, and settled in a 45 °C water bath for 4 h. Then the effect of flocculants on simulated ASP flooding produced water was studied.

## 3. Results and Discussion

### 3.1. Demulsifier

#### 3.1.1. Screening of Demulsifier Single Agent

Table 2 shows the effect of demulsifiers on simulated ASP flooding produced water. It was found that most water-soluble demulsifiers did not decrease the oil concentration compared with that in the reference sample, to which demulsifiers were not added, but the oil-soluble demulsifiers decreased the oil concentration obviously. So, the effect of oil-soluble demulsifiers on simulated ASP flooding-produced water is significantly better than that of water-soluble demulsifiers. Among them, two demulsifiers (DYRPR-1625 and DYRPR-1870) were the most effective; these were thus used in the preparation of mixed demulsifiers. It is obvious that DYRPR-1625 had the best demulsifying efficiency when its concentration was 100 mg/L. The oil concentration of produced water after demulsification was 61.91 mg/L and the oil removal rate could be as high as 96.9%. However, when the concentration of the demulsifier was 200 mg/L, DYRPR-1870 had the best demulsifying efficiency. The oil concentration of produced water was 55.37 mg/L and the oil removal rate reached as high as 97.2%. Although the demulsifying efficiency of DYRPR-1870 was better than that of DYRPR-1625, its advantage was not significant. Therefore, DYRPR-1625 with a concentration of 100 mg/L should be selected as it is more economic.

**Table 2.** Effect of different demulsifiers on alkaline–surfactant–polymer (ASP) flooding-produced water.

| Number | Demulsifier | Demulsifier Concentration (mg/L) | Oil Concentration (mg/L) | Demulsifier Concentration (mg/L) | Oil Concentration (mg/L) |
|---|---|---|---|---|---|
| 0 | Blank sample | 0 | 450.24 | 0 | 450.24 |
| 1 | DPR-1193 | 100 | 396.72 | 200 | 309.32 |
| 2 | DPR-1612 | 100 | 366.29 | 200 | 278.18 |
| 3 | DPR-1814 | 100 | 406.55 | 200 | 321.78 |
| 4 | DPR-1815 | 100 | 372.08 | 200 | 284.25 |
| 5 | DPR-1884 | 100 | 296.75 | 200 | 201.64 |
| 6 | DPR-1885 | 100 | 280.26 | 200 | 189.31 |
| 7 | DYRPR-1625 | 100 | 61.91 | 200 | 77.62 |
| 8 | DYRPR-1712 | 100 | 126.75 | 200 | 114.82 |
| 9 | DYRPR-1881 | 100 | 243.52 | 200 | 174.99 |
| 10 | DYRPR-1882 | 100 | 102.65 | 200 | 63.48 |
| 11 | DYRPR-1869 | 100 | 95.49 | 200 | 57.88 |
| 12 | DYRPR-1870 | 100 | 81.08 | 200 | 55.37 |

### 3.1.2. Compound Demulsifiers

The compounding of two or more demulsifiers is one of the methods to develop effective demulsifiers, taking advantage of the synergistic effect between demulsifiers. Hence, the compounding of two single-agent demulsifiers (DPR-1870 and DYRPR-1625) with the best demulsifying efficiencies was carried out. The combination scheme and treatment effect are shown in Table 3. As shown in Table 3, after compounding, the demulsifying efficiency of the demulsifier was enhanced, the oil concentration of the produced water decreased and the oil removal rate increased obviously. Meanwhile, when the ratio of DYRPR-1625 to DYRPR-1870 was 3:1, it was found that the compound demulsifier had the best treatment effect, and in this condition the compound was named ZY demulsifier. The demulsifiers used in the following experiments were ZY demulsifier.

Then as seen in Figure 1, the oil droplets in the simulated ASP flooding produced water remained basically unchanged without the ZY demulsifier after 4 h settling. When ZY demulsifier was added to the simulated produced water, the size of oil droplets increased significantly after 4 h settling. The result indicates that the oil droplets in the simulated produced water have become unstable after adding ZY demulsifier.

**Table 3.** Different combination schemes and demulsifying effects of compound demulsifiers.

| DYRPR-1625 (mg/L) | DYRPR-1870 (mg/L) | Ratio | Total (mg/L) | Oil Concentration (mg/L) |
|---|---|---|---|---|
| 60 | 60 | 1:1 | 120 | 55.23 |
| 40 | 80 | 1:2 | 120 | 58.46 |
| 30 | 90 | 1:3 | 120 | 60.17 |
| 80 | 40 | 2:1 | 120 | 50.18 |
| 48 | 72 | 2:3 | 120 | 53.68 |
| 90 | 30 | 3:1 | 120 | 48.21 |
| 72 | 48 | 3:2 | 120 | 51.84 |
| 120 | 0 | 1:0 | 120 | 55.67 |
| 0 | 120 | 0:1 | 120 | 64.59 |

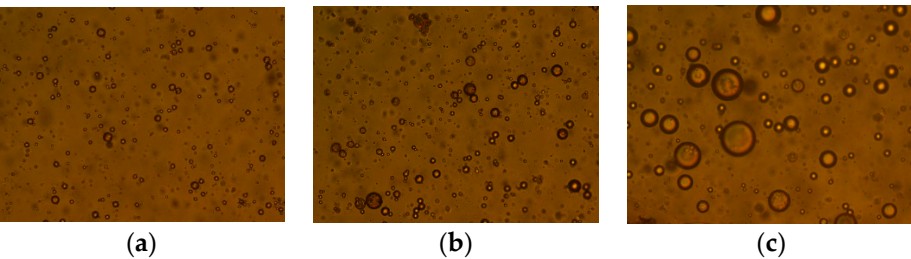

(**a**)          (**b**)          (**c**)

**Figure 1.** Effect of ZY demulsifier on oil droplets. (**a**) At 0 min settling with no ZY demulsifier; (**b**) At 4 h settling with no ZY demulsifier; (**c**) At 4 h settling with ZY demulsifier.

### 3.1.3. Optimization of Demulsifying Conditions

The dosage, settling time and settling temperature of demulsifiers affect the demulsifying efficiency. Therefore, these three conditions of demulsifiers were optimized by experiments.

#### Effect of the Dosage

When the settling temperature was 45 °C and the settling time was 4 h, a specific amount of ZY demulsifier was added to simulated ASP flooding produced water and the mixture was stirred with a stirrer at 200 rpm for 2 min. The effect of the dosage of ZY demulsifier on the oil removal rate was studied. As seen in Figure 2, when the dosage of ZY demulsifier was less than 100 mg/L, with the increase in the dosage, the oil concentration gradually decreased and the oil removal rate gradually increased. When the dosage exceeded 100 mg/L, the oil concentration and oil removal rate tended

to be gentle. Therefore, when the dosage of ZY demulsifier was between 50 mg/L and 100 mg/L, the demulsifying effect was optimal.

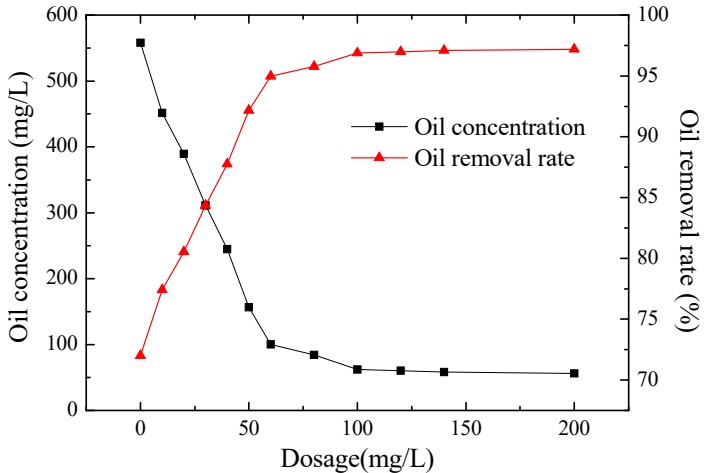

**Figure 2.** Effect of the dosage on oil concentration and oil removal rate.

Effect of the Settling Time

The effect of settling time and oil removal rate of ZY demulsifier on the produced water was studied. The settling temperature was 45 °C, the dosage of ZY demulsifier was 100 mg/L, and the mixture was stirred with a stirrer at 200 rpm for 2 min. The results were compared with those obtained in the absence of the demulsifier under the same conditions.

As seen in Figure 3, the oil concentration of the simulated ASP flooding produced water decreased with the increase in settling time. At the same settling time, 100 mg/L of ZY demulsifier was added to the simulated ASP flooding produced water, and the oil concentration was much less than that in the absence of the demulsifier. After 60 min of settlement, the oil concentration in the absence of the demulsifier decreased to 812 mg/L. Meanwhile, the oil concentration of produced water with ZY demulsifier decreased to 351 mg/L, and the decrement of the oil concentration of simulated ASP flooding-produced water treated with the demulsifier was much larger than that without the demulsifier. When the settling time was 4 h, the oil concentration with ZY demulsifier could be reduced to 61.91 mg/L. At this time, the decline of the oil concentration became very gentle, so the optimal settling time was considered to be 4 h.

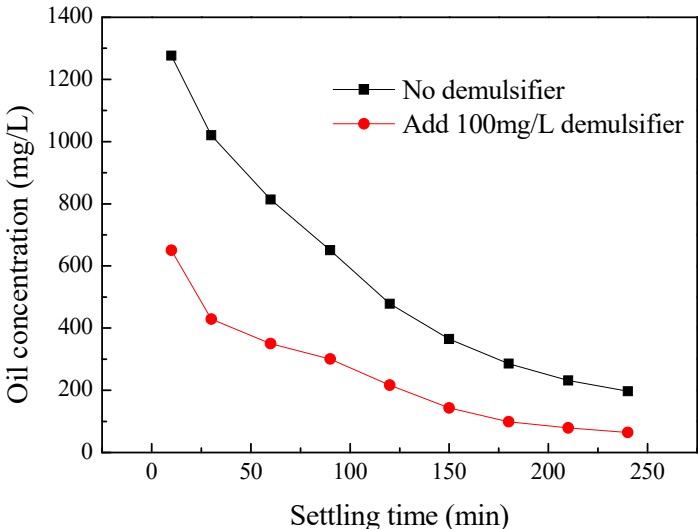

**Figure 3.** Influence of settling time on oil concentration.

Effect of the Settling Temperature

When the dosage of ZY demulsifier was 100 mg/L, stirred with a stirrer at 200 rpm for 2 min, and allowed to settle for 4 h, the effect of the settling temperature on the oil removal rate was studied.

As seen in Figure 4, the oil concentration of the produced water decreased with the increase in the settling temperature. The oil removal rate increased with the increase in the settling temperature. When the settling temperature was raised from 10 °C to 40 °C, the oil concentration of the produced water decreased greatly. As the temperature continued to increase, the oil concentration of the produced water decreased gently and the system gradually reached a steady state. This is because the increase in temperature can speed up the movement of demulsifier molecules, so that more demulsifiers can be analyzed and absorbed to the surface of oil droplets, reducing the stability of the interfacial film. Meanwhile, the increase in temperature can also increase the collision times of oil droplets, increasing the coalescence probability of oil droplets and facilitating the separation of oil and water. Because the temperature of the Daqing oilfield reservoir is 45 °C, the settling temperature in this study was set at 45 °C.

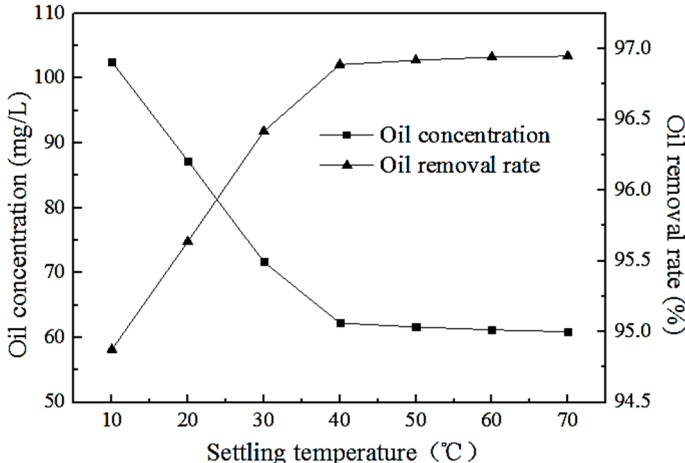

**Figure 4.** Influence of settling temperature on oil concentration and oil removal rate.

Based on the above experimental research, the optimal treatment conditions for ZY demulsifier were determined: the dosage was 100 mg/L, the settling time was 4 h, and the settling temperature was 45 °C. The oil concentration of simulated ASP flooding-produced water could be reduced from the initial 2000 mg/L to 61.91 mg/L and the oil removal rate could reach 96.91%. However, the oil concentration still cannot reach the standard of "double 20" for formation reinjection. Therefore, flocculation treatment of produced water was carried out to study the treatment effects of different flocculants.

*3.2. Flocculants*

3.2.1. Screening of Flocculant Single Agents

Table 4 shows the effect of flocculants on simulated ASP flooding produced water. It was found that the effect organic flocculants on the simulated produced water was significantly better than that of inorganic flocculants. Among them, two flocculants (CPAM and D2N-1650) were the most effective and were thus used in the preparation of compound flocculants. It is obvious that CPAM had the best flocculation efficiency; the oil concentration of produced water after flocculation with this flocculant was 156.32 mg/L.

**Table 4.** Treatment efficiencies of different flocculants on produced water.

| Number | Flocculants | Flocculant Dosage (mg/L) | Oil Concentration (mg/L) |
|--------|-------------|--------------------------|--------------------------|
| 0 | Blank sample | 0 | 450.24 |
| 1 | PAC | 200 | 250.53 |
| 2 | PFS | 200 | 274.03 |
| 3 | PAFC | 200 | 220.41 |
| 4 | D2N-1647 | 40 | 210.63 |
| 5 | D2N-1651 | 40 | 208.79 |
| 6 | D2N-1650 | 40 | 192.23 |
| 7 | D2N-1648 | 40 | 196.31 |
| 8 | CPAM | 40 | 156.32 |
| 9 | APAM | 40 | 214.36 |
| 10 | PAM | 40 | 205.24 |

3.2.2. Compound Flocculants

The use of a synergistic effect between flocculants to compound two or more flocculants is one of the methods for developing high-efficiency agents. Two single agents (CPAM and D2N-1650) with good treatment effects were combined. The compounding scheme and treatment efficiency are shown in Table 5. It can be seen from Table 5 that the treatment efficiency of the flocculants was enhanced after compounding; the oil concentration of the produced water decreased and the oil removal rate increased. Moreover, it was found that when the ratio of CPAM to D2N-1650 was 3:1, the compound flocculant had the best treatment effect. Hence, in this paper the compound flocculant of CPAM and D2N-1650 with a ratio of 3:1 is termed SW flocculant. The flocculant used in subsequent experiments was SW flocculant.

Then as seen in Figure 5, the oil droplets in the simulated ASP flooding produced water remained basically unchanged without SW flocculant after 4 h settling. When SW flocculant was added to the simulated produced water, the size of oil droplets increased after 4 h settling. The result indicates that the oil droplets in the simulated produced water have become unstable after adding SW flocculant. However, the efficiency of SW flocculant is not as good as that of ZY demulsifier.

**Table 5.** Combination schemes and flocculation efficiency of different single-agent.

| CPAM (mg/L) | D2N-1650 (mg/L) | Ratio | Total (mg/L) | Oil concentration (mg/L) |
|-------------|-----------------|-------|--------------|--------------------------|
| 30 | 30 | 1:1 | 60 | 55.23 |
| 20 | 40 | 1:2 | 60 | 58.46 |
| 15 | 45 | 1:3 | 60 | 60.17 |
| 40 | 20 | 2:1 | 60 | 50.18 |
| 24 | 36 | 2:3 | 60 | 53.68 |
| 45 | 15 | 3:1 | 60 | 48.21 |
| 36 | 24 | 3:2 | 60 | 51.84 |
| 48 | 0 | 1:0 | 60 | 55.67 |
| 0 | 48 | 0:1 | 60 | 64.59 |

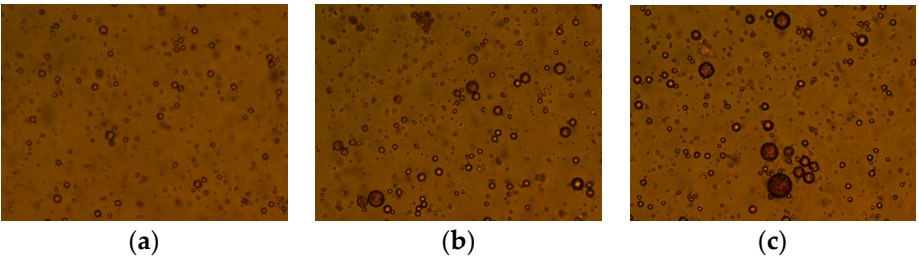

(a)      (b)      (c)

**Figure 5.** Effect of SW flocculant on oil droplets. (**a**) At 0 h settling with no SW flocculant; (**b**) At 4 h settling with no SW flocculant; (**c**) At 4 h settling with SW flocculant.

### 3.2.3. Optimization of Flocculation Conditions

The dosage, stirring intensity, and stirring time of the flocculant affect the flocculation efficiency. Therefore, these three conditions were optimized by experiments.

Effect of the Dosage

The effect of the dosage of SW flocculant on the oil concentration and oil removal rate in produced water was studied, with stirring for 2 min at 200 rpm stirring intensity. Then the mixture was allowed to settle for 4 h at 45 °C. As seen in Figure 6, when the dosage of SW flocculant was less than 60 mg/L, the oil concentration gradually decreased with the increase in the dosage of SW flocculant and the oil removal rate gradually increased. When the dosage of SW flocculant was 60 mg/L, the oil concentration of produced water was reduced to the lowest value, 98.25 mg/L, and the oil removal rate reached the maximum value, 95.08%. Then, with the increase in the dosage of SW flocculant, the oil concentration increased and the oil removal rate decreased. This is because the agent CPAM in the SW flocculant is positively charged. When the concentration of CPAM decreases, on the one hand, the CPAM molecules can be bridged and adsorbed simultaneously with a plurality of negatively charged oil beads. On the other hand, the CPAM can be adsorbed on the oil–water interface by electrostatic interaction, replacing the original position of the surfactant, making the interfacial film arrangement loose and reducing the mechanical strength of the interfacial film. At the same time, surfactants on the oil–water interface were adsorbed on CPAM molecular chains via hydrogen bonds, electrostatic attraction, and hydrophobic action, which reduced the active substances on the stable interface and ultimately led to the decrease in the oil concentration in the water. As the dosage of SW continued to increase, the negative charge on the surface of the oil droplets was completely neutralized and the surface of the oil droplets began to have a positive charge, thereby increasing the repulsive force between the oil droplets, increasing the viscosity of the water phase, and enhancing the steric hindrance of the interface film, which is not conducive to the aggregation and coalescence of the oil beads [18]. The best flocculation effect can only be achieved when the dosage of SW is appropriate. Therefore, the optimum dosage of SW was found to be 60 mg/L.

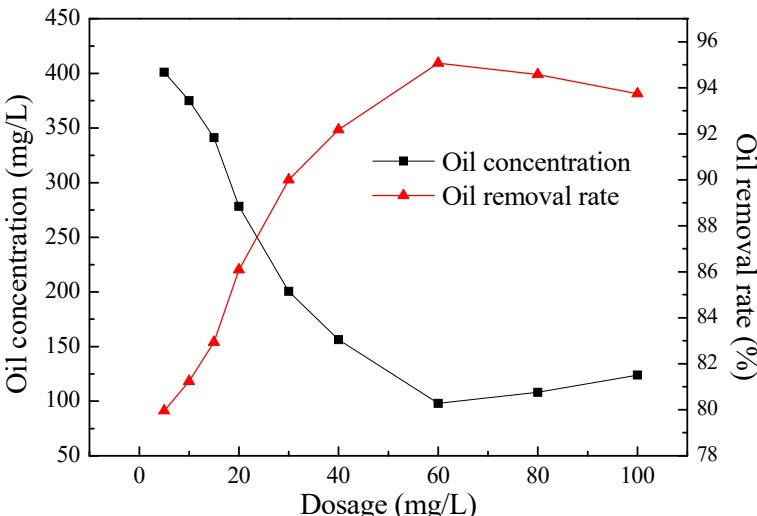

**Figure 6.** Effect of the dosage of flocculant on oil concentration and oil removal rate.

Effect of the Stirring Intensity

With a set dosage of SW of 60 mg/L, the influence of the stirring intensity on the oil removal rate was investigated. It can be seen from Figure 7 that the oil removal rate first increased and then decreased with the increase in the stirring intensity. This is because when the stirring intensity is low, the oil droplets in the water cannot fully collide, flocculate, and settle. However, if the stirring

intensity is too high, the large flocs formed in the water will be broken and re-dispersed in the produced water without subsidence, thus affecting the treatment efficiency. So, as can be seen from Figure 7, the optimum stirring intensity was 200 rpm, at which the oil removal rate reached a maximum value of 95.08%.

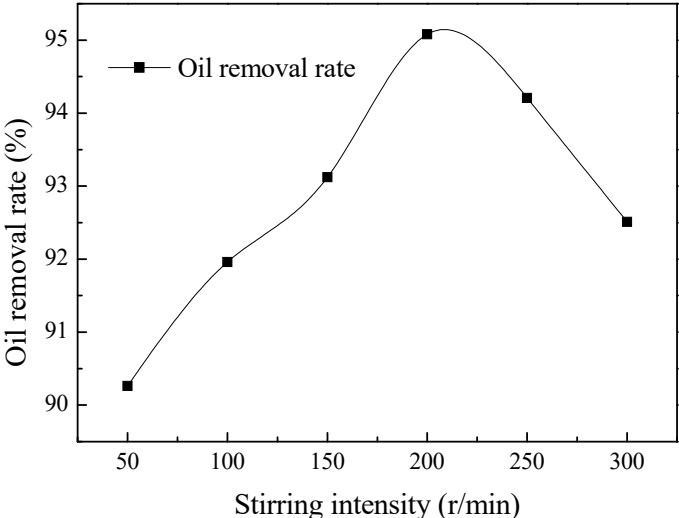

**Figure 7.** Effect of stirring intensity on oil removal rate.

Effect of the Stirring Time

The influence of the stirring time on the oil removal rate was investigated. As can be seen from Figure 8, the oil removal rate increased first and then decreased with the increase in the stirring time. This is because when the stirring time is too short, the emulsified oil particles in the water cannot reach sufficient flocculation. When the stirring time is too long, although the reaction is more sufficient, the large-sized flocs which have been formed are broken and re-dispersed into water and cannot subside, thereby affecting the treatment efficiency. So, as can be seen from Figure 6, the optimum stirring time was found to be 3 min, at which point the oil removal rate reached a maximum value of 95.52%.

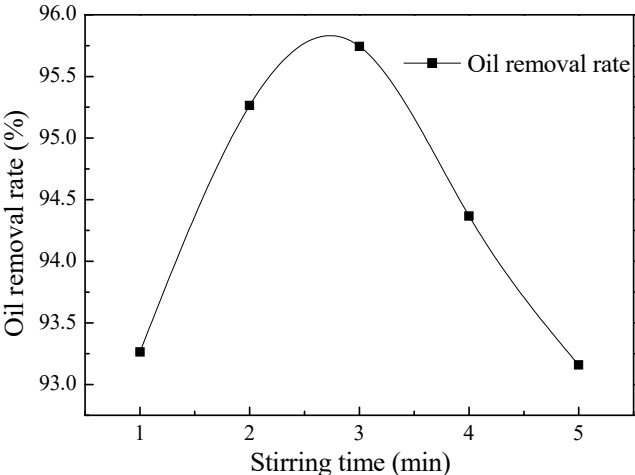

**Figure 8.** Effect of stirring time on oil removal rate.

Based on the above experimental research, SW flocculant as the optimal flocculant is preferred. Meanwhile, the optimal flocculation conditions were obtained: the dosage of SW was 60 mg/L, the stirring intensity was 200 rpm, and the stirring time was 3 min. The oil concentration of simulated ASP flooding-produced water was reduced from the initial 2000 mg/L to 89.6 mg/L and the oil removal

rate reached 95.52%. However, the oil concentration still cannot reach the standard of "double 20" of formation reinjection.

### 3.3. Experimental Study on Demulsification-Flocculation

It was found in the above experimental studies that the simulated ASP flooding-produced water cannot be sufficiently treated with demulsifiers or flocculants alone to reach the standard of formation return water injection. Therefore, the method of demulsification treatment followed by flocculation treatment was considered. The treatment of simulated ASP flooding-produced water by the demulsification–flocculation method was studied, as described below.

#### 3.3.1. Optimization of Demulsification-Flocculation Conditions

The oil concentration in water is an important parameter to evaluate the emulsion stability. According to the analysis above, the main factors influencing the oil concentration were: demulsifier dosage (factor A), flocculant dosage (factor B), settling time (factor C), stirring intensity (factor D), and stirring time (factor E). An $OA_{16}$ ($4^5$) matrix, which is an orthogonal array of five factors and four levels, was employed to assign the considered factors and levels, as shown in Table 6.

**Table 6.** Experimental factors of demulsification–flocculation compound system.

| Level | Experimental Factor | | | | |
| | Demulsifier Dosage (mg/L) A | Flocculant Dosage (mg/L) B | Settling Time (min) C | Stirring Intensity (r/min) D | Stirring Time (min) E |
|---|---|---|---|---|---|
| 1 | 80 | 40 | 180 | 100 | 2 |
| 2 | 90 | 50 | 210 | 150 | 3 |
| 3 | 100 | 60 | 240 | 200 | 4 |
| 4 | 110 | 70 | 270 | 250 | 5 |

The experiment was conducted to assess the five factors and four levels, without considering the interaction between various factors. An orthogonal array experimental design was selected, as shown in Table 7. Sixteen trials were carried out according to the $OA_{16}$ matrix to complete the optimization process. The oil concentration was used to characterize the emulsion stability. In this orthogonal array, the oil concentration was used as the index. The smaller the oil concentration, the better the effect of the demulsification–flocculation method.

**Table 7.** Orthogonal array experimental design.

| Trial No. | Experimental Factor | | | | | Trial No. | Experimental Factor | | | | |
| | A | B | C | D | E | | A | B | C | D | E |
|---|---|---|---|---|---|---|---|---|---|---|---|
| 1 | 1 | 1 | 1 | 1 | 1 | 9 | 3 | 1 | 3 | 4 | 2 |
| 2 | 1 | 2 | 2 | 2 | 2 | 10 | 3 | 2 | 4 | 3 | 1 |
| 3 | 1 | 3 | 3 | 3 | 3 | 11 | 3 | 3 | 1 | 2 | 4 |
| 4 | 1 | 4 | 4 | 4 | 4 | 12 | 3 | 4 | 2 | 1 | 3 |
| 5 | 2 | 1 | 2 | 3 | 4 | 13 | 4 | 1 | 4 | 2 | 3 |
| 6 | 2 | 2 | 1 | 4 | 3 | 14 | 4 | 2 | 3 | 1 | 4 |
| 7 | 2 | 3 | 4 | 1 | 2 | 15 | 4 | 3 | 2 | 4 | 1 |
| 8 | 2 | 4 | 3 | 2 | 1 | 16 | 4 | 4 | 1 | 3 | 2 |

#### 3.3.2. Analysis of the Results of Demulsification-Flocculation

The results of the orthogonal array and range analysis are shown in Table 8. There are two important parameters in a range analysis: $\overline{k_{ji}}$ and $R_j$. $\overline{k_{ji}}$ ($j$, $j$ = A, B, C, D, E and $i$, $i$ = 1, 2, 3, 4) is used to determine the optimal level and the optimal combination of factors. The optimal level for each factor

is obtained when parameter $\overline{k_{ji}}$ is the largest. The range value ($R_j$) indicates the significance of the factor's effect; a larger $R_j$ means that the factor has a bigger impact on the oil concentration.

**Table 8.** Range analysis of the demulsification–flocculation compound system.

| Number | Experimental Factor | | | | | Oil Concentration (mg/L) |
|---|---|---|---|---|---|---|
| | A | B | C | D | E | |
| 1 | 1 | 1 | 1 | 1 | 1 | 38.45 |
| 2 | 1 | 2 | 2 | 2 | 2 | 36.57 |
| 3 | 1 | 3 | 3 | 3 | 3 | 32.62 |
| 4 | 1 | 4 | 4 | 4 | 4 | 34.21 |
| 5 | 2 | 1 | 2 | 3 | 4 | 28.63 |
| 6 | 2 | 2 | 1 | 4 | 3 | 31.63 |
| 7 | 2 | 3 | 4 | 1 | 2 | 28.15 |
| 8 | 2 | 4 | 3 | 2 | 1 | 27.59 |
| 9 | 3 | 1 | 3 | 4 | 2 | 22.19 |
| 10 | 3 | 2 | 4 | 3 | 1 | 21.38 |
| 11 | 3 | 3 | 1 | 2 | 4 | 20.53 |
| 12 | 3 | 4 | 2 | 1 | 3 | 18.96 |
| 13 | 4 | 1 | 4 | 2 | 3 | 20.61 |
| 14 | 4 | 2 | 3 | 1 | 4 | 18.27 |
| 15 | 4 | 3 | 2 | 4 | 1 | 16.59 |
| 16 | 4 | 4 | 1 | 3 | 2 | 19.19 |
| $\overline{k_1}$ | 35.463 | 27.470 | 27.450 | 25.957 | 26.003 | |
| $\overline{k_2}$ | 29.000 | 26.962 | 25.188 | 26.324 | 26.525 | |
| $\overline{k_3}$ | 20.765 | 27.473 | 25.167 | 25.455 | 25.955 | $\Sigma = 415.27$ |
| $\overline{k_4}$ | 18.665 | 24.987 | 26.087 | 26.155 | 25.410 | |
| Optimal levels | $A_4$ | $B_4$ | $C_3$ | $D_3$ | $E_4$ | |
| $R_j$ | 16.798 | 2.997 | 2.283 | 0.870 | 1.115 | |
| Primary and secondary order | ABCED | | | | | |

According to the results in Table 8, the graph of the range values corresponding to each factor are drawn. As can be seen from Figure 9, compared with the range values of different factors ($R_j$), the factors' levels of significance were as follows: demulsifier dosage > flocculant dosage > settling time > stirring time > stirring intensity.

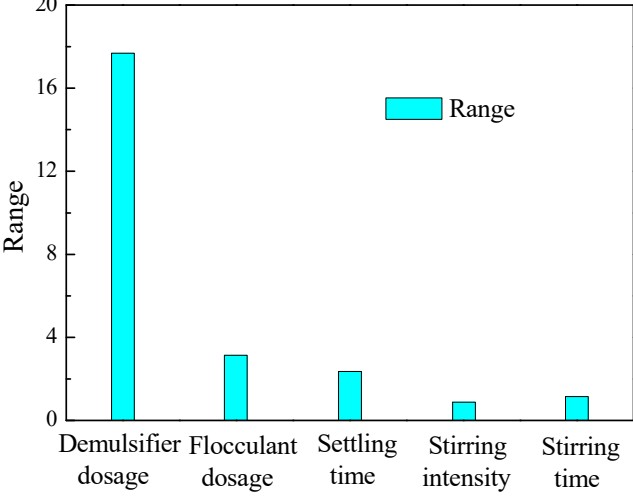

**Figure 9.** Range values of different factors.

The mean values of each factor are shown in Figure 10. Based on the change in the mean value of each factor, it can be observed that the oil concentration sharply decreased from 35.463 mg/L to 18.665 mg/L under optimal conditions, with the dosage of the demusifier increasing from 80 mg/L to 110 mg/L. Concerning the flocculant dosage, the oil concentration reached a minimum of 24.987 mg/L when the dosage of the flocculant was 70 mg/L. Furthermore, the oil concentration decreased as the settling time increased from 180 min to 210 min, then it increased as the settling time increased further. So, when settling time was 210 min, the oil concentration reached a minimum of 25.188 mg/L. The stirring intensity and stirring time were found to have little effect on the oil concentration; the minimum oil concentrations were 25.455 mg/L and 25.410 mg/L, respectively.

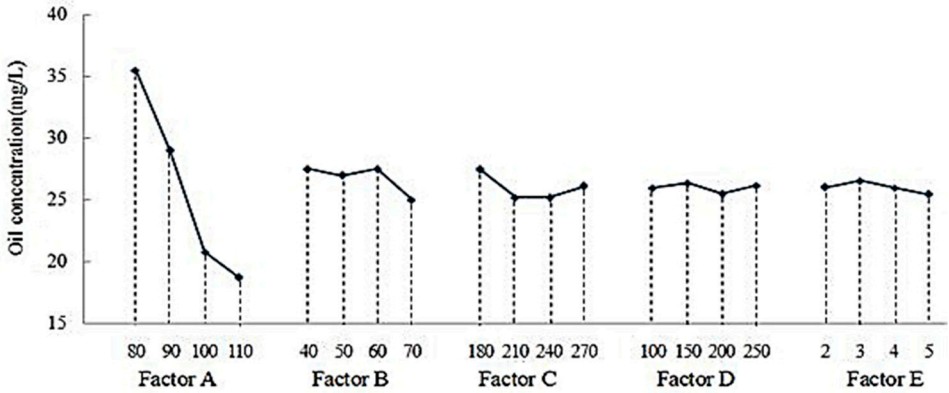

**Figure 10.** Relationship between the five factors and oil concentration.

Therefore, the optimal horizontal condition of factor A is $A_4$, the optimal horizontal condition of factor B is $B_4$, the optimal horizontal condition of factor C is $C_3$, the optimal horizontal condition of factor D is $D_3$, and the optimal horizontal condition of factor E is $E_4$. Through orthogonal experiment, the optimal demulsification–flocculation treatment conditions were optimized as follows: $A_4B_4C_3D_3E_4$—that is, the dosage of the demulsifier was 110 mg/L, the dosage of the flocculant was 70 mg/L, the settling time was 240 min, the stirring intensity was 200 rpm, and the stirring time was 5 min.

### 3.3.3. Verification Analysis of Optimal Treatment Conditions

Since there is no $A_4B_4C_3D_3E_4$ scheme in the orthogonal array, the experiment was repeated under the optimal process conditions to confirm that they truly optimized the oil concentration and treatment efficiency. The processing result of the $A_4B_4C_3D_3E_4$ scheme was compared with those obtained under "direct viewing" and single-factor optimal conditions to determine the optimal processing conditions. As can be seen from Table 7, the optimal treatment scheme was found to be $A_4B_3C_2D_4E_1$. Furthermore, according to the experimental results in demulsifiers and flocculants, the single-factor optimal scheme was found to be $A_3B_3C_3D_3E_2$. The same experimental steps were used to confirm the three experimental schemes.

The processing results of these three experimental schemes are shown in Figure 11. As can be seen from Figure 11, the remaining oil concentration after the treatment of each scheme was in the following order: $A_4B_3C_2D_4E_1$ scheme > $A_3B_3C_3D_3E_2$ scheme > $A_4B_4C_3D_3E_4$ scheme. Figure 11 shows that $A_4B_4C_3D_3E_4$, which was optimized by the orthogonal experiment, had the best treatment efficiency. Therefore, the optimal treatment conditions were: the dosage of the demulsifier was 110 mg/L, the dosage of the flocculant was 70 mg/L, the settling time was 240 min, the stirring intensity was 200 rpm, and the stirring time was 5 min.

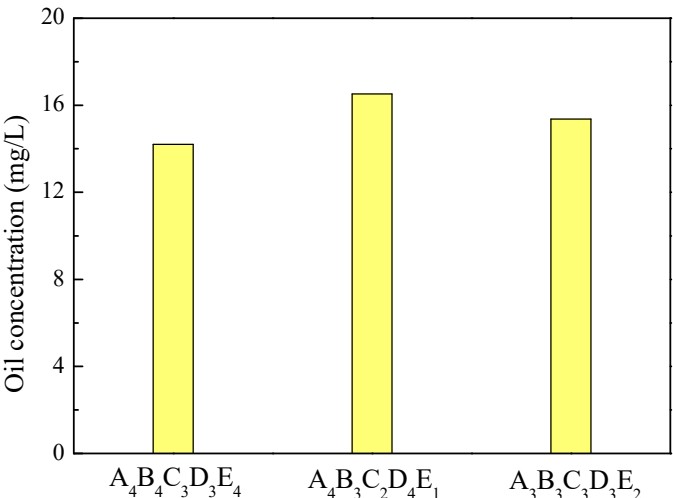

**Figure 11.** Oil concentrations achieved by different treatment schemes.

## 4. Conclusions

Due to the addition of alkali, surfactant, and polymer, the emulsification properties of ASP flooding produced water is extremely stable. Therefore, large amounts of treating agents are used in the treatment process of the produced water, which greatly increases the treatment cost. Therefore, it is very important to provide a method that screening and optimization of chemical reagent for the chemical treatment to deal with the ASP flooding produced water, so that the ASP flooding produced water can be effectively treated.

Demulsification and flocculation tests were carried out on ASP flooding produced water in Daqing Oilfield. Two of the best demulsifier single agents (DYRPR-1625 and DYRPR-1870) and flocculant single agents (CPAM and D2N-1650) were selected for use in our experiments. The optimized demulsifiers and flocculants were compounded separately to obtain the optimal demulsifier ZY and the optimal flocculant SW, and the best ratios were determined to be DYRPR-1625: DYRPR-1870 = 3:1 and CPAM: D2N-1650 = 2:1, respectively.

It is difficult to enable the simulated ASP flooding produced water to reach the standard of reinjection only by using demulsifier or flocculant treatments. Therefore, the optimized demulsifiers and flocculants were compounded and the simulated ASP flooding produced water was subjected to demulsification–flocculation treatments, assessed by orthogonal experiments. The results showed that the optimal treatment conditions were as follows: the dosage of the demulsifier was 110 mg/L, the dosage of the flocculant was 70 mg/L, the settling time was 240 min, the stirring intensity was 200 rpm, and the stirring time was 5 min. Under these optimal conditions, the oil content in the produced water was greatly reduced, thus reducing the treatment cost.

**Author Contributions:** B.H. and J.W. put forward the idea of the experiments in this paper and wrote the paper; W.Z. and C.F. designed a series of experiments scheme; Y.W. and X.L. contributed to the results analysis. All authors reviewed the manuscript.

**Funding:** This work was financially supported by National Natural Science Foundation of China (51804077); Postdoctoral researchers settled in Heilongjiang research funding, Study on the influence mechanism of clay particles on the stability of produced water from ASP flooding, LBH-Q17035; Outstanding scientific research talents of northeast petroleum university, SJQHB201803.

**Conflicts of Interest:** The authors declare no conflict of interest.

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
