# Peer review of "Screening and Optimization of Demulsifiers and Flocculants Based on ASP Flooding-Produced Water"

_processes, doi:10.3390/pr7040239_

Round 1

Reviewer 1 Report

There is an ecological request that the process water of ASP flooding must be purified before returning it into the process circuit or into nature. The authors describe a combination of demulsification and flocculation of the contaminations. A screening of the main factors for water purification is described and their significances evaluated.

This is an interesting paper, not only for people involved in petroleum production. It is nice systematic and deserves publication. I only recommend a small addition: “With ASP flooding a methodology of alkaline–surfactant–polymer flooding is meant.”

Author Response

Response to Reviewer 1 Comments:

First of all, thank you very much for your comments on this article. We tried our best to improve the manuscript according to your comments and made some changes in the manuscript. And the replies to your comments are as follows. We appreciate for your warm work earnestly, and hope that the correction will meet with approval. Once again, thank you very much for your comments and suggestions.

Point 1: This is an interesting paper, not only for people involved in petroleum production. It is nice systematic and deserves publication. I only recommend a small addition: “With ASP flooding a methodology of alkaline–surfactant–polymer flooding is meant.”

Response 1: Special thanks to the problems which you proposed. We are deeply sorry that we regret the mean of ASP flooding has not been described. We have made correction according to your comments in paper, line 33.

We have tried our best to revise our manuscript according to the comments. We would like to express our great appreciation to you for comments on our paper. Looking forward to hearing from you.

Thank you and best regards.

Reviewer 2 Report

Application of some flocculants and demulsifiers on oil/ water mixture has been studied. Please see my comments as follows:

Line 89: The aim of the study and it significance should be explained at the end of the Introduction

Line 92, "No 1 Oil Production" should be better specified

Line 103, I have never seen this term "mineral water",  do you mean synthetic water?? I suggest the composition of the prepared water to be provided in a Table.

2.4 and 2.5, You need to specify the supplier of reagents (flocculants and demulsifiers), and add more technical information, for example, the molecular weight of the flocculant, their purity etc

I suggest adding some photo of the emulsion, before and after treatment for clarification

line 384, not clear

Line 392, "the produced water", is it the same water as you prepared? why using a different term here?

Author Response

Response to Reviewer 2 Comments:

First of all, thank you very much for your comments on this article. We tried our best to improve the manuscript according to your comments and made some changes in the manuscript. And the replies to your comments are as follows. We appreciate for your warm work earnestly, and hope that the correction will meet with approval. Once again, thank you very much for your comments and suggestions.

Point 1: Line 89: The aim of the study and it significance should be explained at the end of the Introduction

Response 1: Special thanks to the problems which you proposed. We are deeply sorry that the purpose and significance of the study are not sufficiently detailed. It is our negligence and we are sorry about this. We have made correction according to the Reviewer's comments from line 87 to line 89, the content as follow: "By studying the screening and optimization method of demulsification and flocculation for the treatment of ASP flooding produced water, we aim to provide a method that screening and optimization of chemical reagent for the chemical treatment to deal with the ASP flooding produced water, so that the ASP flooding produced water can be effectively treated."

Point 2: Line 92, "No 1 Oil Production" should be better specified

Response 2: Special thanks to the problems which you proposed. "No 1 Oil Production" is a company which producing oil in Daqing, called "No.1 Oil Production Company". The oil in paper was provided by this company.

Point 3: Line 103, I have never seen this term "mineral water", do you mean synthetic water?? I suggest the composition of the prepared water to be provided in a Table.

Response 3: Special thanks to the problems which you proposed. Mineral water means simulated formation water. Experiments in study need to use mineral water to prepare simulated produced water. In addition after research, we found that "mineralized water" was more accurate in the description, so the mineral water in paper was all changed to mineralized water. In order to make the reader understand more clearly, we have made correction according to your comments. In paper, the composition of mineralized water is provided in a table.

Point 4: 2.4 and 2.5, You need to specify the supplier of reagents (flocculants and demulsifiers), and add more technical information, for example, the molecular weight of the flocculant, their purity etc.

Response 4: Special thanks to the problems which you proposed. We added the the supplier and purity of reagents according to your comments in paper.

Point 5: I suggest adding some photo of the emulsion, before and after treatment for clarification

Response 5: Special thanks to the problems which you proposed. We have adding some photo of the emulsion according to your comments in paper, which can be seen at page 5 and page 9.

Point 6: line 384, not clear

Response 6: Special thanks to the problems which you proposed. Line 384, we would like to express the meaning and purpose of screening and optimization of demulsifiers and flocculants. However the meaning which we expressed is not clear. It is our negligence and we are sorry about this. According to your comments in paper, we have made correction.

Point 7: Line 392, "the produced water", is it the same water as you prepared? why using a different term here?

Response 7: Special thanks to the problems which you proposed. Line 392, "the produced water" is the same water as we prepared. It is our negligence to not writing "simulated ASP flooding produced water ", and we are sorry about this. So we have made correction according to your comments in paper.

We have tried our best to revise our manuscript according to the comments. Attached please find the revised version, which we would like to submit for your kind consideration. We would like to express our great appreciation to you for comments on our paper. Looking forward to hearing from you.

Thank you and best regards.

Round 2

Reviewer 2 Report

The authors have adequately responded to my comments.